# Positional Behavior of Introduced Monk Parakeets (*Myiopsitta monachus*) in an Urban Landscape

**DOI:** 10.3390/ani12182372

**Published:** 2022-09-11

**Authors:** Michael C. Granatosky, Melody W. Young, Victoria Herr, Chloe Chai, Anisa Raidah, Judy Njeri Kairo, Adaeze Anaekwe, Allison Havens, Bettina Zou, Billy Ding, Christopher Chen, David De Leon, Harshal Shah, Jordan Valentin, Lucas Hildreth, Taylor Castro, Timothy Li, Andy Yeung, Edwin Dickinson, Dionisios Youlatos

**Affiliations:** 1Department of Anatomy, College of Osteopathic Medicine, New York Institute of Technology, Old Westbury, NY 11568, USA; 2Center for Biomedical Innovation, College of Osteopathic Medicine, New York Institute of Technology, Old Westbury, NY 11568, USA; 3College of Osteopathic Medicine, New York Institute of Technology, Old Westbury, NY 11568, USA; 4Department of Zoology, School of Biology, Aristotle University of Thessaloniki, GR-54124 Thessaloniki, Greece

**Keywords:** arboreal, locomotion, posture, Psittaciformes, Psittacidae, urban, exotic species, introduced species

## Abstract

**Simple Summary:**

Positional behaviors comprise the entirety of animals’ locomotion and posture. Often, these positional behaviors are paired with information about sußbstrate characteristics (e.g., orientation, diameter, texture, height) and frequency to gain an ecological perspective of when and why an animal utilizes a particular behavior. Thus far, quantitative studies of positional behavior have been limited to mammals, leaving a major gap in our understanding of how animals utilize their environment. In this study, we present the first quantitative report of positional behavior within Aves, presenting scan sampling data from an established colony of Monk Parakeets (*Myiopsitta monachus*) from Brooklyn, New York City. Parrots exhibited a strong preference for small and terminal branches when perching arboreally. Such a pattern is consistent with arboreal primates. We also observed an increase in locomotor diversity on artificial versus naturally occurring substrates. This demonstrates the potential importance of a flexible behavioral repertoire in facilitating a successful transition towards an urban landscape in introduced species and underscores the need for further studies exploring positional behaviors among urban wildlife.

**Abstract:**

Positional behaviors have been broadly quantified across the Order Primates, and in several other mammalian lineages, to contextualize adaptations to, and evolution within, an arboreal environment. Outside of Mammalia, however, such data are yet to be reported. In this study, we present the first quantitative report of positional behavior within Aves, presenting 11,246 observations of scan sampling data from a colony of Monk Parakeets (*Myiopsitta monachus*) from Brooklyn, New York City. Each scan recorded locomotor and postural behavior and information about weather condition, temperature, and substrate properties (e.g., type, size, orientation). A distinction was also recorded between natural and artificial substrates. Parrots exhibited a strong preference for small and terminal branches, a selection which may reflect targeted foraging of new fruit growth and leaf-buds. We further observed that the gait transition from walking to sidling appears primarily driven by substrate size, with the former preferred on the ground and on large, broad substrates and the latter used to navigate smaller branches. Finally, we observed an increase in locomotor diversity on artificial versus naturally occurring substrates. This demonstrates the importance of a flexible behavioral repertoire in facilitating a successful transition towards an urban landscape in introduced species.

## 1. Introduction

The use of arboreal substrates presents many challenges, reflecting the difficulties of navigating a complex three-dimensional environment using substrates of varying orientation, size, texture, compliance, and gap-distance [1,2,3,4]. Understanding how arboreal animals successfully negotiate such conditions has been the focus of numerous field- [5,6,7,8] and laboratory-based [9,10,11,12,13] studies for two primary reasons. First, it is well-established that humans derive from arboreal primates, and unravelling the selective pressures of our ancestors contextualizes our own evolutionary history [1,14,15]. Second, anatomically, modern humans are not well-adapted to arboreal life. Even individuals that regularly ascend trees for sustenance [16] perform poorly when compared to non-human primates. While the fundamental biomechanics of arboreal locomotion are well-established, there remain critical shortcomings to our analyses of arboreal behaviors, most notably with regards to taxonomic sampling of which, with limited exceptions [7,8,17,18,19], has been dominated by observational studies of primates (see [20] for a review). This sampling bias raises the question of whether observed behaviors associated with arboreal locomotion are indicative of tetrapods in general, mammals, or just primates.

Arboreal locomotion has evolved numerous times across tetrapods, and some lineages have become highly specialized for life in the trees [3,21,22,23,24,25]. Among birds, the parrots (Order: Psittaciformes) are a primarily arboreal lineage [26,27,28] that have evolved numerous anatomical features well-known to be associated with arboreal locomotion [29,30,31]. Briefly, these include long zygodactylous digits for grasping around supports [30,32], a distal elongation of the penultimate phalanx associated with a shortening of the proximal phalanges [32,33,34,35] that has been posited to increase grasping force [32,36], high mobility at the hip joint, and digital and tarsometatarsal pads that are textured and highly sensitive [37]. Further, Psittaciformes tend to have relatively short tarsometatarsi compared to other avian species [31]. This morphology serves to reduce limb length, allowing the animal to maintain closer contact to the substrate, thereby decreasing the gravitational pitching moment during climbing and rolling torques on thin arboreal substrates [38]. Parrots have also co-opted the feeding system and neck musculature to function as a third limb during climbing, an exaptation completely unique to the order [37,39]. Despite the well-known arboreal tendencies of parrots, there remains limited information about the way these animals exploit their environment in terms of locomotion and postures. Previous studies have been largely anecdotal in nature [40,41], and classifications of their locomotion and posture have been grossly categorical (e.g., “arboreal” and “perching”) [30,34].

Positional behaviors comprise the entirety of animals’ locomotion (i.e., behaviors that involve movement of the center of mass) and posture (i.e., behaviors that involve no movement of the center of mass) [42]. Often, these positional behaviors are paired with information about substrate characteristics (e.g., orientation, diameter, texture, height) and frequency to gain an ecological perspective of when and why an animal utilizes a particular behavior [5,6,42]. Traditionally, positional behaviors serve a corollary role in studies of evolutionary morphology. Studies of this sort usually proceed by noting salient differences in behavior and morphology in two or more species, establishing form/function relationships between the two, and attempting to explain the relationships using arguments derived from biomechanical principles [8,43,44,45]. Success in such an endeavor depends on the quality of both morphological and behavioral datasets. As such, the lack of detailed information hampers efforts to assess the evolutionary importance of behaviors and to construct realistic locomotor groupings or classifications [46]. More recently, gaining an understanding of positional behavior has been critical for conservation efforts, as such information is essential for focusing on the protection of critical habitats and for establishing naturalistic environments in zoological institutions and rehabilitation facilities [47]. Taken together, positional behavior is therefore an important component of the behavioral repertoire of a species.

This study represents the first analysis of positional behavior in wild parrots. Our study population comprises an introduced population of Monk Parakeets (*Myiopsitta monachus*) established in Brooklyn, New York [48]. Any study of positional behavior is inherently descriptive in nature and produces more data that can reasonably be discussed in a single manuscript. As such, we have developed the following major predictions to focus the investigation: Based on morphological studies [30,32,34], we expect perching to be the dominant posture regardless of substrate use (Prediction 1). In terms of locomotion, we anticipate that, like most birds, powered flight is common [49,50,51]. However, when moving in trees, we predict overall locomotor diversity to increase [20], including substantial use of beak-assisted climbing and descent [4,39,40,41]. We expect hopping behaviors to be entirely absent based on proposed neuromuscular constraints in parrots [31] (Prediction 2). Based on parallels made between primates and parrots [52] and considerations about force-generating capabilities of grasping in birds [32,53,54], we expect small-diameter, horizontally oriented branches will be preferred, and that movement on the ground and on large-diameter substrates will be infrequent (Prediction 3). As our study focuses upon an introduced species within in urban context [48,55,56], we also address how adaptation to their newfound urban landscape influences locomotor behavior by comparing levels of locomotor diversity. As the range of locomotor behaviors has been shown to increase in the trees [20], we expect locomotor diversity in Monk Parakeets to be higher when moving on arboreal compared to terrestrial substrates. Further, as introduced species are known for their locomotor flexibility [57,58,59], we expect locomotor diversity in Monk Parakeets to be higher when moving on artificial versus natural substrates (Prediction 4).

## 2. Materials and Methods

### 2.1. Study Species

Monk Parakeets (Psittacidae: *Myiopsitta monachus*), are small (body length of 30 cm, wingspan of 48 cm, and body mass of 100 g), brightly colored green parrots with a greyish breast, greenish-yellow abdomen, long tail feathers, and zygodactylous feet [30,60]. Like most Psittaciformes, they are long-lived, with estimates ranging anywhere from 20 to 30 years [48,60]. Monk Parakeets usually feed on a variety of seeds, fruits, blossoms, leaf buds, tree parts, grasses, and insects [48,61,62]. Native to temperate and subtropical regions of Argentina and surrounding countries, the Monk Parakeets have become one of the most successful introduced species [55]. Self-sustaining introduced populations have become established throughout Europe and the United States [56,63,64]. The species adapts well to urban areas and has established populations in large cities, such as Miami, Chicago, and New York [48,55,61]. Part of the species’ ability to withstand harsh temperate conditions has been attributed to their nest-building behavior. Unlike most parrots that nest in tree hollows, Monk Parakeets build large stick-nests that house large communal colonies. The external heat radiating off each individual paired with the insulating properties of the nest allows the animals to maintain a safe core body temperature, even in freezing conditions [55,56,61,63,65].

### 2.2. Study Site

All data were collected from Green-Wood Cemetery (25th Street, Brooklyn, NY, 11232; N 40.65811, W 73.99460), which is a 193.4 ha cemetery in the western portion of Brooklyn, New York City. The Monk Parakeets have established a large nest situated among the gothic architecture of the main gate (Figure 1). Current surveys estimate approximately 36–50 individuals at Green-Wood Cemetery [48].

### 2.3. Data Collection

Behavioral observations were conducted whenever possible over 101 days between 17 January and 31 June 2021. The decision to end the study was based on two factors. First, by sampling from January through the end of June, we were able to capture the entire range of temperature extremes usually reported in New York State [66]. Second, the last novel behavior was observed on 21 March 2021, meaning there was an additional three months of sampling where no new behaviors were recorded. As such, we are confident our study captured the entire range of positional diversity of Monk Parakeets at this study site.

We developed an ethogram (Table 1) for this study based on Hunt and colleagues [5], Dilger [40], Brockway [41], and a short (1–15 January 2021) ad libitum sampling session by the principal investigator. We only observed one behavior (i.e., uprighting) not on the original ethogram during the formal sampling period. Prior to the initiation of the study, all observers conducted training sessions (3 in total, consisting of 5–10 investigators each) with the principal investigator to gain familiarity and confidence in behavioral classification and sampling protocols. These training sessions were also used as an opportunity to assess the potential of interobserver error in the sampling protocol (see the Section 2.4 below).

As the Monk Parakeets often travelled in large groups and distinguishing individuals was difficult, we used a five-minute interval scan sampling to collect behavioral data. Each scan lasted less than one minute. We scanned the Monk Parakeets from left to right or in a clockwise sweep to avoid potential bias toward given individuals. We recorded the behavior of all visible individuals during scanning, with no individual sampled twice. Number of individuals (mean ± standard deviation: 5.76 ± 4.32) observed per scan varied considerably. During each scan, we identified the predominant behavior of the sampled individual after observing it for five seconds. We also cataloged weather condition (i.e., sunny, overcast, snow flurries, rain), temperature, latitude and longitude, substrate type, and orientation (Table 2). For short flights or leaps between supports, the substrate was coded based on the landing site. No substrate or orientation code was scored if the animals’ feet were not in contact with the support (e.g., flight, hovering) during the five second within-scan observational period. All observers were equipped with 8 × 21 compact binoculars (BriGenius, Shenzhen, China) to aid in behavioral scoring.

### 2.4. Data Analysis

Following established methods [67,68], we utilized pilot data collected during the training sessions (see above) to assess the possibility of interobserver error in data scoring. After the initial walkthrough of the study goals and an introduction to the ethogram and scan sampling protocols, each observer watched the nest for one hour (i.e., twelve scans) and was asked to score positional behavior, substrate type, and orientation. We used the crosstab function in MATLAB (MathWorks, Inc., Natick, MA) to calculate a χ^2^-statistic to assess whether the likelihood of scoring a particular positional behavior, substrate type, or orientation was influenced by the observer. We detected no significant interobserver variation in scoring with regards to locomotion (all χ^2^-values across three training groups < 45.76; all *p*-values > 0.838), posture (all χ^2^-values across three training groups < 6.95; all *p*-values > 0.642), substrate type (all χ^2^-values across three training groups < 42.88; all *p*-values > 0.537), or orientation (all χ^2^-values across three training groups < 4.78; all *p*-values > 0.936). Based on these results, we were satisfied that interobserver error in the main dataset (i.e., data collected after the training sessions, 101 days between 17 January and 31 June 2021) should be minimal.

To address the main dataset (i.e., data collected after the training sessions, 101 days between 17 January and 31 June 2021), we first divided the observational sample into locomotor and postural datasets. From these, we calculated proportions of different positional behaviors, substrate types, and orientation used by the animals during the sampling period. We used the crosstab function in MATLAB to conduct a series of χ^2^-tests to assess whether observing a particular positional behavior was influenced by substrate type, orientation, month, or weather condition. As weather condition had a low number of observations, we used Yate’s χ^2^-tests for these analyses.

We conducted a second analysis comparing locomotor diversity [20] between arboreal, terrestrial, and artificial substrates. Briefly, this study co-opted the Shannon–Wiener diversity index to calculate a singular measure of locomotor diversity. The Shannon–Wiener diversity index is traditionally used in the ecology literature to provide a statistically comparable metric to quantify the diversity of species composition in a specific area of interest (e.g., forest, transect, etc.) [69,70,71,72]. This study employs this same logic using the proportion of each locomotor behavior observed on a particular substrate to calculate a substrate-specific locomotor diversity index. The locomotor diversity index (LDI) is calculated as:$$\mathrm{Locomotor}\;\mathrm{diversity}\;\mathrm{index}=-\sum p_{i\;}\ln p_{i}$$
where *p_i_* is the proportion of a particular locomotor behavior (e.g., walking, leaping, climbing) out of the total combination of locomotor behaviors (e.g., proportion of walking + proportion of leaping + proportion of climbing). A low LDI indicates that Monk Parakeets frequently use only a few different locomotor behaviors on a particular substrate. High LDI values indicate that Monk Parakeets frequently use many different locomotor behaviors on a particular substrate.

## 3. Results

Across the six-month sampling period, we collected 11,246 observations comprising of 7337 postural (65.24%) and 3909 locomotor scans (34.76%) (Appendix A). Throughout the study period, the temperature ranged from −3.89 to 36.11 °C. Weather conditions varied between snow flurries (*n* = 11 observations, 0.10%), overcast (*n* = 2954 observations, 26.27%), rain (*n* = 539 observations, 4.79%), and sun (*n* = 7742 observations, 68.84%) (Appendix A). In total, twenty observers contributed to the overall sample.

During non-aerial activities (i.e., excluding flight and hovering), Monk Parakeets spent a majority of their time on their large elaborate nest (*n* = 3777 observations, 44.76%) (Figure 2 and Appendix A). During non-aerial activities outside the nest, Monk Parakeets were mainly engaged in arboreal locomotion (*n* = 3424 observations, 40.59%). Animals favored both terminal (*n* = 1670 observations, 19.80%) and small-diameter branches (*n* = 1240 observations, 14.70%), while the use of medium (*n* = 458 observations, 5.43%), large (*n* = 38 observations, 0.45%), and very large branches (*n* = 18 observations, 0.21%) was less common. Terrestrial (*n* = 342 observations, 4.05%) locomotion and posture and movement on artificial structures (*n* = 894 observations, 10.60%) were also observed. During non-aerial locomotion, Monk Parakeets were primarily observed on horizontally (*n* = 5841 observations, 69.24%) and obliquely oriented (*n* = 2163 observations, 25.64%) supports, while the use of vertical supports (*n* = 432 observations, 5.12%) was more limited (Appendix A).

Locomotor behaviors were dominated by flight (*n* = 2639 observations, 67.51%). However, during non-aerial activities (*n* = 1098 observations), short flights within trees (*n* = 432 observations, 39.34%), walking (*n* = 223 observations, 20.31%), and sidling (*n* = 125 observations, 11.38%) were the most common forms of locomotion, while anti-pronograde behaviors, sensu stricto Stern [14], were uncommon (bridging: *n* = 3 observations, 0.27%; suspensory: *n* = 4 observations, 0.36%; uprighting: *n* = 5 observations, 0.45%). Climbing behaviors (pooled Climb, Climb_beak-assisted, and Climb_wing-assisted) accounted for 7.92% of non-aerial behaviors (*n* = 87 observations) and were observed in generally equal proportions (Figure 3 and Appendix A).

The time of year had a significant influence on locomotor behavior (χ^2^ = 504.62; *p* < 0.001), where walking became more common in March, April, May, and June, and leaping with the assistance of the wings, which was a fairly common locomotor behavior early in the study, was not observed after March (Figure 3). As a greater proportion of time was spent on the ground (see above), this was associated with a general increase in walking (Appendix A). Weather also had a significant influence on locomotor behavior (χ^2^ = 252.58; *p* < 0.001), where non-aerial movements were less common during snow flurries and rain (Appendix A).

Substrate had a significant influence on locomotor behavior (Figure 3; χ^2^ = 1517; *p* < 0.001). When on the nest, Monk Parakeets spent a majority of the time making short flights and then quickly landing nearby (*n* = 233 observations, 64.54%). Climbing the nest using only the hindlimbs was also common (*n* = 37 observations, 10.25%). Terrestrial locomotion was dominated by walking (*n* = 169 observations, 91.35%), but hopping (*n* = 11 observations, 5.95%) and running (*n* = 5 observations, 2.70%) were also observed. On all arboreal substrates, excluding those of large diameter, short flights to branches within the same tree were most common. On terminal branches, leaping using the wings was also quite common (*n* = 67 observations, 38.07%). On small and medium branches, sidling is the dominant locomotor mode (small: *n* = 74 observations, 35.24%; medium: *n* = 29 observations, 40.28%). Walking was also a major component of the locomotor repertoire on medium branches (*n* = 12 observations, 16.67%), and the primary form of locomotion on large branches (*n* = 3 observations, 42.86%). Climbing with the assistance of the beak was observed on all arboreal substrates (terminal: *n* = 1 observation, 0.57%; small: *n* = 9 observations, 4.29%; large: *n* = 1 observation, 14.29%; very large: *n* = 4 observations, 50.00%), excluding medium diameter branches. Movement on artificial substrates was variable, with walking being most common (*n* = 25 observations, 31.65%), followed by short flights to nearby substrates (*n* = 15 observations, 18.99%), leaping with the assistance of the wings (*n* = 7 observations, 8.86%), and equal contributions of climbing, climbing with assistance of the beak, climbing with assistance of the wings, and descent with assistance of the beak (*n* = 5 observations, 6.33%). Accordingly, locomotor diversity was greater on artificial supports (LDI = 2.14), compared to arboreal substrates (LDI = 1.78), the nest (LDI 1.35), or the ground (LDI = 0.35).

Substrate orientation significantly influenced the observed locomotor modes (Figure 3; χ^2^ = 597.97; *p* < 0.001). Across all orientations, short flights to nearby supports were common (Appendix A). On horizontal substrates, walking (*n* = 203 observations, 35.74%) and sidling (*n* = 76 observations, 13.38%) comprised the greatest proportion of observed locomotor behaviors. On oblique supports, walking (*n* = 20 observations, 6.29%) and sidling (*n* = 49 observations, 15.41%) were also quite common but leaping with the assistance of the wings (*n* = 75 observations, 23.58%) was the most frequently observed behavior. Excluding short flights to nearby supports, locomotion on vertical supports consisted primarily of climbing (*n* = 37 observations, 17.45%) and climbing with assistance of the beak (*n* = 26 observations, 12.36%).

Postural behaviors were dominated by perching (*n* = 7112 observations, 96.93%; Appendix A). The time of year had a significant influence on postural behavior (χ^2^ = 167.25; *p* < 0.001; Appendix A), where standing increased significantly in May and June. As a greater proportion of time was spent on the ground (see above), this was associated with a general increase in standing. Weather also had a significant influence on postural behavior (χ^2^ = 115.15; *p* < 0.001), where standing and clinging were relatively more common in sunny and rainy conditions, respectively (Figure 4 and Appendix A).

Substrate had a significant influence on postural behavior (χ^2^ = 8123.90; *p* < 0.001). This was largely driven by Monk Parakeets standing when on the ground (*n* = 157 observations, 100.00%) and more frequently clinging to large (*n* = 5 observations, 16.13%) and very large supports (*n* = 6 observations, 60.00%). Similarly, substrate orientation significantly influenced the observed postural modes (χ^2^ = 1076.50; *p* < 0.001). This difference was attributable to a greater proportion of clinging (*n* = 23 observations, 10.50%) and cantilevering (*n* = 14 observations, 6.39%) on vertical substrates (Figure 4).

## 4. Discussion

We have provided the first quantitative assessment of positional behavior in parrots, and, to our knowledge, the first quantitative report of positional behavior outside of Mammalia. Monk Parakeets are arboreal specialists, that when moving in the trees, favor horizontally oriented terminal branch substrates. In this way, it is possible to draw parallels between the positional behaviors of Psittaciformes and arboreal primates [20].

With regards to postural mode, it is safe to say that Monk Parakeets are “perching birds.” Thus, Prediction 1 is supported. The overwhelming use of perching behavior harkens back to Bock and Miller’s [21] argument that the zygodactylous foot configuration is not an adaptation for clinging on vertical substrates, but instead a means to ensure grasping forces produced by the digital flexors are opposed to each other to maximize grip strength around cylindrical objects [30,32]. As the digital flexor muscles of at least some Psittaciformes possess a tendon-locking mechanism [73], Monk Parakeets may be capable of maintaining a strong foothold on the support while expending minimal metabolic energy. The presence of a digital-locking mechanism has not been confirmed in Monk Parakeets [73].

Based on our definitions of support size, we anticipated that Monk Parakeets would primarily utilize small-diameter substrates, as these would allow the foot to completely wrap around the substrate with no overlap between the toes and theoretically maximize grip force potential [53]. While the use of small-diameter supports was common, terminal branches were the most commonly used substrates by Monk Parakeets. Thus, Prediction 3 is partially supported. As discussed thoroughly in the primate literature, terminal branches allow animals new foraging opportunities as these are the primary site for new fruit growth and leaf-buds [1,74,75], which have been shown to be a favorite food item of the Monk Parakeets in Brooklyn [48]. However, movement on terminal branches is dangerous as such slight substrates may oscillate under the weight of the animal, increasing the risk of becoming unbalanced and falling [1,22,76]. While some arboreal species mitigate these concerns by adopting suspensory positional behavior [19,22], such a strategy was rarely observed among Monk Parakeets (Appendix A). Instead, Monk Parakeets, most likely due to their small body mass, like many similarly sized primates [43,77], use their grasping feet to counteract toppling forces and maintain balance on the branch [54,78]. However, the diameter of terminal branches means the toes demonstrate substantial overlap between each other and arguably indicates that the digital flexors are primarily active on the early, ascending portion of a Hill-type length–tension curve [79]. While it is possible that the passive tendon-locking mechanism described above may supersede these considerations, there are insufficient data available to interpret how substrate diameter influences grasping ability in birds. Future work following Sustaita and Hertel [53] and Ward and colleagues [80] would help to address this gap.

Flight was the most observed form of locomotion among Monk Parakeets Even when moving within a tree or around the nest, flight and hovering remained the primary form of movement. Thus, this aspect of Prediction 2 is supported. This finding is not surprising considering the lower cost of transport associated with flight compared to terrestrial locomotion [49,50,51]. While the energetic cost associated with hindlimb-based locomotion in parrots is unknown, it is likely that these costs are quite high. Compared to most other birds, parrots have relatively short tarsometatarsi that reduce the overall length of the limb [31], thereby requiring increased stride frequencies during locomotion [81,82]. Further, like primates, parrots have large feet with considerable mass positioned distally along the hindlimb [83,84]. Higher stride frequency and distal weight distribution are both associated with increased locomotor costs [82,83,84,85]. As such, it seems probable that the locomotor behavior of Monk Parakeets may be explained through considerations of the overall energetic budget. Future work should explore how differences in body size, a factor known to influence locomotor costs, influence the positional behaviors between parrot species and in other birds.

Walking and sidling were the most observed forms of hindlimb-driven locomotion. Both walking and sidling involved the alternating left/right patterning of the limbs. This finding is in accordance with Provini and Höfling [31], who note that parrots violate general patterns in avian locomotion by being a relatively small-bodied arboreal lineage that does not hop. While we did observe Monk Parakeets hop occasionally, this movement was rare (*n* = 28 observations) and most often observed on the ground. Thus, this aspect of Prediction 2 was rejected. The presence of hopping in Monk Parakeets rejects the idea of a possible neuromuscular constraint in the lineage [31] and highlights a gap in our understanding of why certain avian lineages hop while others do not. Further, it is unclear what factors trigger parrots to adopt hopping gaits.

The occurrence of walking and sidling was largely driven by substrate diameter. Generally, walking was observed on terrestrial and large-diameter substrates, while sidling became more common as substrate size decreased. To our knowledge, sidling was first described by Dilger [40], and involves sidewise progression along the perch, where one foot is moved before the other in a shuffling manner. Originally, Dilger [40] proposed that the use of sidling was reserved for slow speed progression, but Brockway [41] noted that sidling was adopted even at high speeds in budgerigars (*Melopsittacus undulatus*). Based on the marked shift between walking and sidling dependent on substrate diameter, a finding in accordance with Young et al. [86] in Rosy-faced Lovebirds (*Agapornis roseicollis*), we propose that the use of sidling represents a solution to an anatomical constraint. Compared to most tetrapod lineages, birds have greatly reduced the number of tarsal and metatarsal bones into a singular tarsometatarsus [87]. Further, articular surfaces with the knee proximally and phalanges distally are characterized with noticeable deep trochleae [28,87]. As such, the ability to generate parasagittal movements (e.g., inversion and eversion) at the distal joints of the hindlimb is limited. By adopting sidling, birds can position their grasping feet perpendicular to the long axis of the substrate, thus ensuring grasping abilities on thin arboreal supports where inversion/eversion is not possible [86].

Contrary to Prediction 2, we did not observe a large proportion of beak-assisted climbing. Parrots are unique among tetrapods as they have co-opted the feeding system and neck musculature into an effective third limb to both propel and power the body during vertical ascent. Climbing with the assistance of the beak comprised 3.28% of substrate-based locomotion (Appendix A) and was largely restricted to movement on vertical supports (Appendix A). This is consistent with recent laboratory experiments that the use of the head as a third limb is only present at very steep inclines (e.g., 67.5°), and does not become ubiquitous until vertical ascent [39]. Monk Parakeets infrequently utilized vertical supports and were rarely observed on very large substrates (e.g., tree trunks). This is in stark contrast to the movements of woodpeckers, nuthatches, and treecreepers [21,29,38,88]. Webster and colleagues [4] suggested that the use of the beak as a third limb is more common among larger-bodied Psittaciformes, but this suggestion is anecdotal in nature and further work on the positional behavior of additional parrot species is required to verify this claim.

### 4.1. Positional Behavior in an Urban Environment

Urban habitats are dramatically modified from their natural state, creating unique challenges and selection pressures for organisms that reside in them. Winchell and colleagues [58] also noted that arboreal species are particularly impacted because the anthropogenic structures with which they interact differ from trees in structural, material, and surface properties. While multiple studies have demonstrated that locomotor behaviors and performance change in response to urban landscapes [59], to our knowledge this is the first dataset reporting how overall positional behavior is influenced. Ideally, we would directly compare the positional behavior of Monk Parakeetsin their natural range in juxtaposition to the current dataset. In the absence of such data, we compared the locomotor behavior of Monk Parakeets on artificial substrates compared to natural substrates (i.e., terrestrial and arboreal substrates and the nest). Monk Parakeets were quite adept at movement on artificial substrates, and were often observed moving on telephone wires, concrete (e.g., building facades), marble (e.g., gravestones), and metal (e.g., vehicles). On these artificial structures, Monk Parakeets adopted a more diverse locomotor repertoire with less reliance on a single locomotor modality. Accordingly, locomotor diversity was higher when moving on artificial structures compared to natural substrates. Thus, Prediction 4 was supported. The data within this study are insufficient to determine why these changes occur between natural and artificial substrates but highlight the importance of behavioral flexibility in the locomotor behavior of urban species. It is unclear whether behavioral flexibility is the response to movement in urban environments, or whether it is an important trait of species able to colonize and persist in urban areas.

### 4.2. Limitations

A study of this nature is faced with a number of limitations that must be addressed. Most notable was the multiple observers that may introduce interobserver error in scoring. While training sessions and subsequent interobserver statistical analyses (see Section 3) suggest these effects to be minimal, such error cannot be removed entirely. This manuscript represents the culmination of student-led research aimed at introducing animal behavior research to a broader range of individuals and increasing inclusivity in the research process. As such, it was not possible to reduce the number of observers without compromising inclusivity and broadening participation in science. Another limitation is in the use of scan sampling, which was required because of the flighty nature of the animals and the inability to differentiate individuals. Scan sampling provides an unbiased assessment of the activity budget of an animal but suffers from missing rare or uncommon behaviors. This is evidenced by the almost exclusive use of perching and flight. However, these limitations do not detract from the overall findings and interpretations of the manuscript, as we identified no new behaviors after the first three months of sampling (see above). Further, as the goal was to capture the range and frequency of certain behaviors, scan sampling was most appropriate [89]. Next, the study started in January. During this month, the observers were just becoming acquainted with the study site and animals. Accordingly, there are fewer observations for January than other months. This means we may have missed information about how Monk Parakeets behave in cold weather conditions. As the ability for these parrots to survive in these conditions is critical for understanding their success as introduced species, we hope future studies will focus more sampling effort on this critical time period. Finally, the most rigorous assessments of positional behavior are able to account for a relative abundance of substrates [19,90]. However, these types of analyses are usually limited to captive settings where substrate conditions can be manipulated or appropriately counted. Such relative abundance analyses were not possible within this study.

## 5. Conclusions

This study revealed several novel and potentially valuable insights into the locomotor repertoire of Monk Parakeets. Firstly, we determined that these birds exhibit a marked preference for small and terminal branches when perching arboreally, a selection which may reflect the ability of the digital flexor musculature to most effectively develop high grip forces and engage a passive tendon-locking mechanism in these tightly curled pedal postures. The preference for small and terminal branches is similar to what is observed in primates. In this way, Monk Parakeets likely demonstrate a myriad of anatomical characteristics similar to primates that allow for movement on such precarious substrates. Further work is required to assess if and how larger parrot species navigate the challenges of terminal branch movement. Secondly, we report that the gait transition from walking to sidling appears primarily driven by substrate size, with the former preferred on the ground and on large, broad substrates and the latter used to navigate smaller branches. Such a gait pattern has been poorly investigated in the literature and likely represents a novel behavioral solution to an anatomical constraint [86]. If so, the presence of sidling is likely present in many avian species, or at least those that spend significant time in the trees [31]. Thirdly, contrary to our initial prediction, climbing with the use of the beak was relatively rare and was primarily influenced by the orientation of the substrate. Perhaps this is not surprising as the use of the beak, at least in Rosy-faced Lovebirds [39], does not become common until orientations >67.5°. Thus far, beak use has only been investigated in small-bodied parrot species (this study, and [40,41,42]) and additional sampling effort across Psittaciformes is required to understand the conditions that elicit tri-pedal gaits in parrots. Finally, we observed an increase in locomotor diversity on artificial versus naturally occurring substrates, a shift which underscores the necessity of behavioral flexibility within introduced urban species to successfully navigate a novel, variable, and often challenging environment.

## Figures and Tables

**Figure 1 animals-12-02372-f001:**
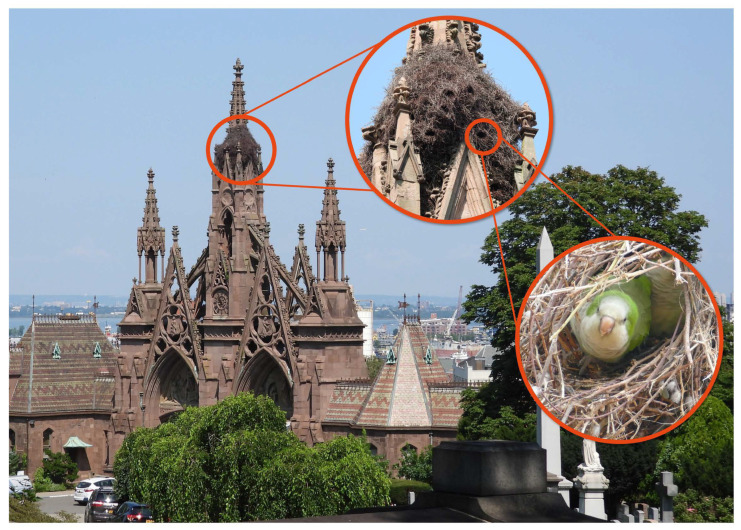
A large population of introduced Monk Parakeets (*Myiopsitta monachus*) have established a nest in Green-Wood Cemetery, Brooklyn, NY.

**Figure 2 animals-12-02372-f002:**
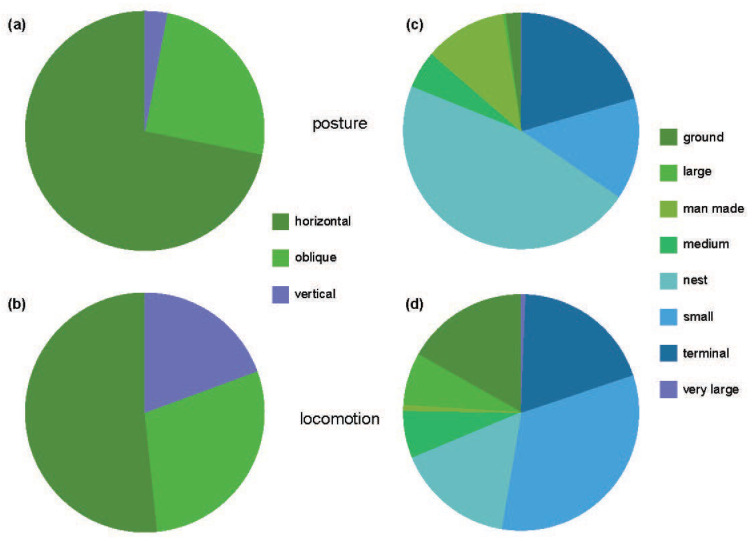
Pie charts demonstrating the relative proportions of substrate orientation (**a**,**b**) and substrate type (**c**,**d**) collected during postural (**a**,**c**) and locomotor behaviors (**b**,**d**) from an established colony of Monk Parakeets (*Myiopsitta monachus*) from Brooklyn, New York City, during the study period.

**Figure 3 animals-12-02372-f003:**
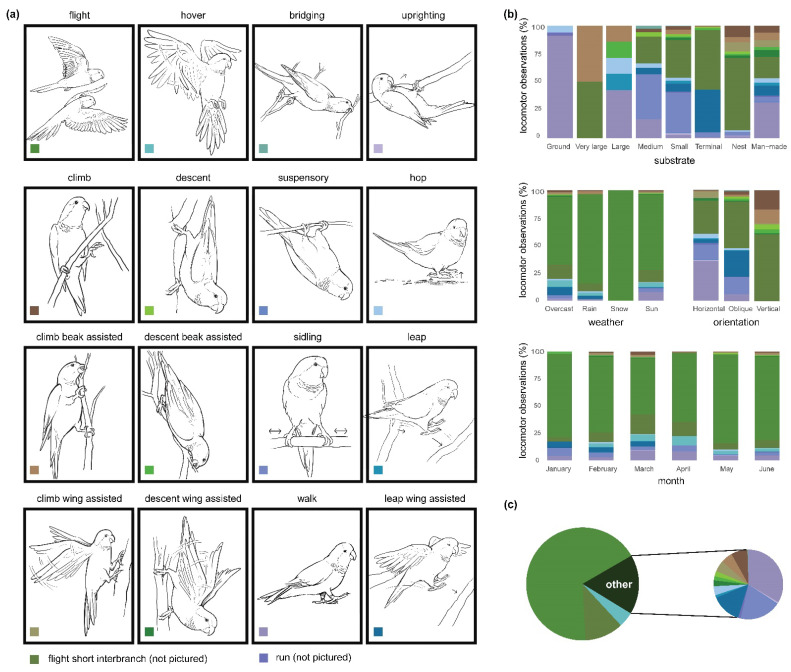
Illustrations and proportions of locomotor behaviors (**a**) observed based on differing substrate types, orientations, weather conditions, and time of year (**b**), collected from an established colony of Monk Parakeets (*Myiopsitta monachus*) from Brooklyn, New York City, during the study period. The pie chart (**c**) illustrates the overall proportions of locomotor behaviors. The expanded pie chart illustrates proportions of non-aerial behaviors.

**Figure 4 animals-12-02372-f004:**
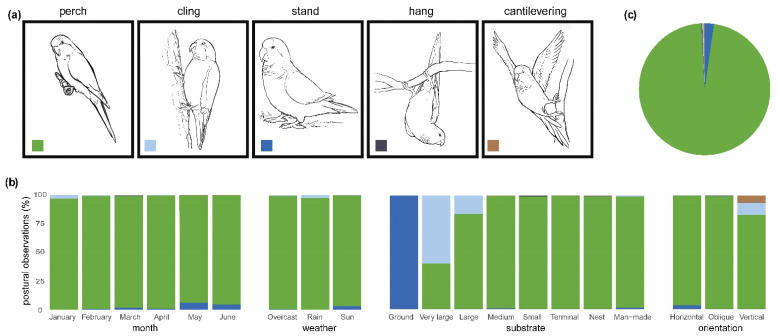
Illustrations and proportions of postural behaviors (**a**) observed based on differing substrate types, orientations, weather conditions, and time of year (**b**), collected from an established colony of Monk Parakeets (*Myiopsitta monachus*) from Brooklyn, New York City, during the study period. The pie chart (**c**) illustrates the overall proportions of postural behaviors.

**Table 1 animals-12-02372-t001:** Ethogram for this study based on Hunt and colleagues [5], Dilger [40], Brockway [41], and a short (1–15 January 2021) ad libitum sampling session by the principal investigator.

Positional Behavior	Description
Locomotion	
Flight	Locomotion in which no body part touches a substrate. The individual’s wings are slightly angled which allows them to deflect the air downward to generate lift. The trunk is held in a pronograde position.
Hover	The wings are extended and flexed rapidly such that flight is achieved without significant movement accomplished in any direction.
Flight_short_interbranch	With a starting position of resting upright on a branch, this is a fast but short locomotion propelled by flapping the wings, which fully supports the body weight as the bird settles on another nearby branch. This behavior is differentiated from Leap_wing-assisted based on the relatively long distances between subsequent branches.
Bridge	A torso pronograde gap-closing movement by anchoring the feet on a substrate while using the beak to reach across a gap to take hold of another substrate. Grip is retained until a secure position is established on the other side, then pulls the body across the open space to the substrate on the other side.
Climb	With an orthograde trunk orientation, forward motion is achieved as each hindlimb asynchronously protracts then retracts, allowing each foot to make alternating contact with the substrate. The weight-bearing is entirely on the hindlimbs without the support of the beak or wings. The feet may be flexed or extended depending on substrate size. The tail may be used as a prop.
Climb_beak-assisted	With an orthograde trunk orientation, upward motion is achieved as each hindlimb asynchronously protracts then retracts, allowing each foot to make alternating contact with the substrate. Simultaneously, the neck is stretched, allowing the beak to grasp a substrate such that the weight-bearing is on the hindlimbs and the beak. After upward movement is achieved, the neck retracts and the beak releases its grasp of the substrate. The feet may be flexed or extended depending on the substrate size. The tail may be used as a prop.
Climb_wing-assisted	With an orthograde trunk orientation, upward motion is achieved as each hindlimb asynchronously protracts then retracts, allowing each foot to make alternating contact with the substrate. Simultaneously, the wings are flapped. The torso is held pronograde to the substrate. The feet may be flexed or extended depending on the substrate size. The tail may be used as a prop.
Climb_wingbeak-assisted	With an orthograde trunk orientation, upward motion is achieved as each hindlimb asynchronously protracts then retracts, allowing each foot to make alternating contact with the substrate. Simultaneously, the neck is stretched, allowing the beak to grasp a substrate, and at least one wing is flapped. After upward movement is achieved, the neck retracts and the beak releases its grasp of the substrate. The feet may be flexed or extended depending on the substrate size. The tail may be used as a prop.
Descent	With an orthograde trunk orientation, a head-first downward movement is achieved as each hindlimb asynchronously protracts then retracts, allowing each foot to make alternating contact with the substrate.
Descent_beak-assisted	With an orthograde trunk orientation, a head-first downward movement on the substrate is achieved as each hindlimb asynchronously protracts then retracts, allowing each foot to make alternating contact with the substrate. Simultaneously, the beak grasps the substrate such that the weight-bearing is on the beak and the hindlimbs. After downward movement is achieved, the neck retracts and the beak releases its grasp of the substrate.
Descent_wing-assisted	With an orthograde trunk orientation, a head-first downward movement is achieved as each hindlimb asynchronously protracts then retracts, allowing each foot to make alternating contact with the substrate. Simultaneously, the wings are flapped.
Descent_wingbeak-assisted	With an orthograde trunk orientation, downward motion is achieved as each hindlimb asynchronously protracts then retracts, allowing each foot to make alternating contact with the substrate. Simultaneously, the neck is stretched, allowing the beak to grasp a substrate such that the weight-bearing is on the hindlimbs and the beak. During this time, at least one wing is flapping. After downward movement is achieved, the neck retracts and the beak releases its grasp of the substrate.
Hop	Bipedal locomotion, wherein the feet push off and land almost simultaneously on the substrate. The majority of the weight-bearing is on the hindlimbs, with no substantial support from any other body part. There is a period of free flight. As the individual contacts the substrate, the feet are in an extended position. The trunk is held horizontal.
Leap	A gap-crossing locomotion with a pronograde trunk orientation that primarily uses the hindlimbs to thrust forward. The hindlimbs and back are initially in flexed position and then are vigorously extended. There is a period of free flight until the hindlimbs land on the substrate. The wings are not involved.
Leap_wing-assisted	A gap-crossing locomotion that primarily uses the hindlimbs to thrust forward. The flexed hindlimbs and back are forcefully extended with the assistance of the wings. There is an extended period of free flight, until the individual grasps and lands on the substrate. The trunk is held in a pronograde position throughout the locomotion. This behavior is differentiated from Flight_short_interbranch based on the relatively short distances between subsequent branches.
Run	With a pronograde trunk orientation, forward motion is achieved as each hindlimb asynchronously protracts then retracts, allowing each foot to make alternating contact with the substrate. The weight-bearing is entirely on the hindlimbs without the support of another body part. The feet may be flexed or extended depending on the substrate size.
Sidling	Involves sidewise progression along the perch, where one foot is moved before the other in a shuffling manner.
Suspensory	Locomotion on a substrate using the hindlimbs, which are anchored around a substrate to support the full body weight, which is in an inverted position and moving along the substrate.
Uprighting	From a hanging position, where the hindlimbs are anchored on a substrate with the body inverted, the bird uses its beak to propel its body weight over the branch and into an upright position.
Walk	With a pronograde trunk orientation, forward motion is achieved as each hindlimb asynchronously protracts then retracts, allowing each foot to make alternating contact with the substrate. The weight-bearing is entirely on the hindlimbs without the support of another body part. The feet may be flexed or extended depending on the substrate size.
Posture	
Cantilevering	A postural position on a stable substrate with a near-vertical plane. Both feet are anchored and grasped firmly on the substrate as the trunk is held rigid and horizontal, and then the subject extends and reaches out. This posture is maintained for several seconds.
Cling	A postural position on a substrate with a near-vertical plane. The feet are extended and claws are responsible for anchoring to the substrate and bearing most of the weight, with no significant support from other body parts as its trunk is held in a near-vertical orientation.
Hang	A postural position on a substrate with a near-horizontal plane. The animal is inverted. Most of the body weight is supported by the feet grasping the substrate above the subject’s center of mass. The trunk may be pronograde or orthograde as the feet firmly grasp the substrate.
Perch	Sitting upright and resting on a substrate with the hindlimbs flexed and phalanges grasping around the substrate.
Stand	A postural position on a horizontal substrate. The hindlimbs support most of the weight, with no significant support from other body parts. The feet are splayed in an extended position and a pronograde trunk orientation.

**Table 2 animals-12-02372-t002:** Definition and description of substrate conditions.

Substrate Variable	Definition
Substrate type	
Ground	Ground and related substrates (e.g., rocks, roots, logs)
Very large	Arboreal substrates with a diameter > 20 cm (e.g., tree trunks)
Large	Arboreal substrates with a diameter larger than the dorsoventral height of the animal but < 20 cm (e.g., tree boughs)
Medium	Arboreal substrates with a diameter approximately equal to the dorsoventral height of the animal
Small	Arboreal substrates with a diameter approximately equal to foot span (i.e., complete coverage by the hallux and foredigits)
Terminal	Arboreal substrates with a diameter less than foot span (i.e., overlapping coverage by the hallux and foredigits)
Nest	Communal stick nest
Artificial	Man-made substrates (e.g., telephone wires and poles, building facades, roofs)
Orientation	
Horizontal	Angle between 0° and 22.5°
Oblique	Angle between 22.5° and 67.5°
Vertical	Angle between 67.5° and 90°

## Data Availability

All data can be accessed at: https://docs.google.com/spreadsheets/d/1HqfQkVkUvvvcuM_Ez6W2IKUAvT5oKkYWqHvhEFbZZjo/edit#gid=0 (accessed on 14 July 2022).

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
