# Peer review of "Positional Behavior of Introduced Monk Parakeets (Myiopsitta monachus) in an Urban Landscape"

_animals, 2022, doi:10.3390/ani12182372_

Round 1
Reviewer 1 Report
This manuscript presents a comprehensive ethogram and quantitative analysis of positional and locomotor behaviors of wild Quaker parrots in New York. Although much more common in the mammal behavior literature – particularly primates – studies such as this one, examining how birds use different substrates – are far less typical, and this work provides a refreshing and rather unique perspective in this regard. The authors used standard focal and scan sampling techniques to generate a wealth of data over a 101-day period encompassing a variety of weather conditions. In addition to quantifying the frequencies of substrate types, body positions and orientations, use of the feet, beak, and wings, and locomotor modes, the authors cleverly utilized the Shannon index to calculate and compare the ‘diversity’ of locomotor behaviors on artificial versus natural substrates. The authors found that postural behaviors were dominated by perching and that terminal branches were most commonly used, although substrate type significantly influenced locomotor behavior. Slightly unexpected (for parrots) was that flight was the most common locomotor mode used, and the frequencies of beak-assisted climbing were surprisingly low. The authors also found that ‘sidling’ was the preferred method for moving along small-diameter substrates, and that artificial substrates elicited the greatest diversity of locomotor behaviors.
I feel that the manuscript is succinct and very well written. There were lots of results, but they were well organized and presented with an effective narrative flow. I found no fatal flaws in the design or execution, and any weaknesses in these were were well confronted in the discussion. I feel that the manuscript makes a substantive contribution to our understanding of “arboreal” behavior in non-mammalian tetrapods, as well as to the discipline of urban (behavioral) ecology. I'm afraid that I do not have any comments, because the authors have fully addressed my concerns from a previous version of this manuscript that I had reviewed for a different journal.
Author Response
Thank you so much for the kind review. We are pleased that we addressed all your concerns in an earlier version of this manuscript. Thank you for your time.
Reviewer 2 Report
I am quite impressed with this analysis of parrot behavior and believe it represents a unique contribution to the scientific literature. In particular, the ethogram drawings (Figure 4) are excellent and have not been presented previously. I also find the comparison between sets of postures and weather conditions, substrates, and seasonality interesting. The manuscript is complete and tells a good story.
I would request clarification or modification related to the following:
1. The repeated statement that the study represents the first of its kind in a non-mammalian species (Lines 21-24, 31-34, 103-104, 498-499) should be walked back some. It may be true that the paper provides a unique combination of positional behavior descriptions with an analysis of the ecological context in which each behavior is displayed, but that hardly makes the paper the first of its kind to present an ethogram for a bird species and then describe the conditions under which certain behaviors are most likely to be exhibited. A search of literature using the terms “ethogram” and “birds” results in numerous hits dating back several decades for a variety of avian species. Many of the papers I found present ethograms alone. Others group behaviors into general categories and then analyze their occurrence quantitatively. I did not find anything in the literature precisely like what is presented in these authors’ contribution; however, there is a long history in ornithology of comparing positional behavior to the animals’ social and structural environments that is dismissed when the authors say they are first to think of doing this kind of analysis. I would encourage the authors to be more specific about how their work is different from what has been done in the past.
2. I think the analysis that compares bird positional behavior to human versus natural substrate is interesting. This kind of thing was called for by Burgio et al. (2020) in their species account of the Monk Parakeet. See: Burgio, K. R., C. B. van Rees, K. E. Block, P. Pyle, M. A. Patten, M. F. Spreyer, and E. H. Bucher (2020). Monk Parakeet (Myiopsitta monachus), version 1.0. In Birds of the World (P. G. Rodewald, Editor). Cornell Lab of Ornithology, Ithaca, NY, USA. https://fanyv88.com:443/https/doi.org/10.2173/bow.monpar.01
3. I would recommend the authors use the more common name of this species, the Monk Parakeet, rather than the Quaker Parrot. The use of “Quaker Parrot” is not wrong, but in checking a variety of international sources (ICUN, International Ornithological Committee, Global Invasive Species Database), the use of Monk Parakeet is more commonplace.
4. There are six references to the species being an “invasive species” and four references to it being an “introduced species.” These phrases have different definitions. Invasive species are non-native species that cause ecological or economic harm, whereas introduced species are simply non-native. According to the Global Invasive Species Database, the Monk Parakeet in North America is not causing any known ecological or economic harm, so I think references to it being “invasive” should be deleted or modified. Indeed, the authors make a big point in the Introduction section about how important it is to study an invasive species, but then never follow-up in the Discussion section. For that reason, I don’t think the “invasive” angle or literary hook is needed. The authors may still want to make the point that the bird is introduced and lives only in urban areas outside of its natural range.
5. It is not clear why there is a large difference in samples between postural (n= 7,337) and locomotor scans (n=3,909). The Methods section implies the two kinds of information were collected together during 5-minute scans of a flock. Could the author explain this in Methods within the passage in Lines 186-197?
6. One of the assumptions of a valid Chi-square test is that 80% of expected values must be greater than 5 in all cells of the contingency table. I am doubtful that the test comparing locomotion to weather conditions because there were only 11 observations of snow. This would make the expected values in that whole column smaller than 5, and that would represent more than 20% of the cell total. I believe the authors can fix this by combining the rain and snow categories and running the analysis again. They will likely get the same results as before, but it is still important not to violate test assumptions.
7. There is an extra word, “Thus” at the end of Line 322 that can be deleted.
Author Response
We thank the reviewer for their suggestions. We have made almost all of them. These have greatly improved the quality of the manuscript.

Reviewer 3 Report
Dear Authors
It was with great pleasure that I read this manuscript. The article is well-written, in language that even non-specialists can understand, and contains new and interesting information about the behaviorism of Quaker parrots. Many threads and interesting observations are included in the manuscript, so it would be good to complete the Conclusions section and improve the quality of Figures 3b and 4b before publishing this work.
Regards
Author Response
Thank you for the kind review, we have since expanded the conclusions section to better reflect the large amount of data presented in this paper. We appreciate the sugggestion and believe it has greatly improved the quality of the work. As for the Figures 3-4, we have high resolution PDFs that have been submitted to the editor. The embedded files in the manuscripts were lower quality screenshots. Thank you for your time.
Reviewer 4 Report
The manuscript presents a quantitative analysis of positional and postural behaviors of Quaker parrots living in an urban habitat. While similar studies have been conducted on mammals, this is the first study focusing on birds in an urban environment. The results are thus interesting and novel. The manuscript is generally well written, the presentation of the results might need some improvements. More detailed comments below.
Introduction
Title: I would change the title in “Positional behaviour of introduced Quaker parrots in an urban landscape (or habitat)”
Line 56-57: awkward wording, perhaps change “awful moving at heights” with “not well-adapted to arboreal life”
Line 57-59: fall from heights in workplaces depends on a multitude of other factors and the fact that humans are not well-suited to arboreal life is likely to be the least important of these. I would delete this sentence and only focus on your main point, i.e. adaptation to life in the tree top.
Line 64: missing a [ before [21]
Line 91-94: it is not clear what these correlations are and that do they show. Please, consider rephrasing this sentence and make it clearer for all readers
Line 96: delete “on behaviour”
Line 101-102: it is certainly important but I would tone it down a notch, perhaps “Positional behaviour is therefore an important component of the behavioural repertoire of a species”
Line 113: add “a” in “in single manuscript”
Line 112-115: I would break this sentence in two to improve readability: Any study of positional behavior is inherently descriptive in nature and produces more data that can reasonably be discussed in a single manuscript. We have therefore developed the following major predictions based on relevant background information and to focus investigation:
Line 114-115: relevant background information on what and from where? And to focus investigation on what?
Line 119: change “to be” with “is”
Line 128-129: it is not clear how your expectations of higher diversity in arboreal vs terrestrial and artificial vs natural substrates are telling anything about the adaptation to urban life. You focus only on urban parrots, which means you can surely learn more about their behaviour within an urban habitat. However, to truly understand the effect of urbanisation on a specific behaviour you would need to compare urban with non-urban populations, which is not what you did.
Methods
Line 166: were the observations equally spread in time (for example conducted at 2 or 3 days interval) or rather conducted when possible?
Table 1: the description of flight can be shortened as it include some redundant info (sentences 2-3). In the description of “Climb” change in “The tail may be used as a prop”. Same for all other descriptions.
Lines 187-191: how many individuals were observed during each scan (mean + range)? I also think you should state if the observation were preferentially conducted during good weather conditions or not, looks like you had fewer observations in January (thus colder month) compared to the other months.
Line 213: I would change “principle” with “main” dataset
Line 220: the way this sentence is written sounds like you conducted X2-tests on proportions, rather than raw numbers. I would rephrase it slightly to make this clearer.
Results
Tables: there are way too many tables here, it is very difficult to follow the flow of the manuscript. In addition, most of the info is duplicated in the text. Please consider reporting some of the Tables in an alternate graphic form easier to read (for example only graphing behaviours >2%) or perhaps include some of these as supplementary material
Line 242-245: these numbers are a repetition of what is already presented in Table 4, here is only useful to know that you encountered all possible weather conditions.
Table 3: I would order the behaviours according to their proportion in decreasing order of proportion, makes it easier to read through the table
Line 250: delete “observed in these conditions”
Line 256-263: none of these numbers (besides medium arboreal substrate) match those reported in Table 5, or at least is not clear how the authors calculated these numbers if different from those presented in the table.
Line 263-266: I am afraid I cannot see what the authors are claiming here as Table 6 shows the positional behaviours in relation to months, not use of substrate vs month. Also, there are very few observations in January compared to the other months, so how this influences the result? Also, according to the number of observations in Table 6, most observations were collected in March. Is this a result of higher behavioural activity or a sampling bias? See my previous comment about the timing of data collection
Line 266-269: again, this is just repetition of what is reported in Table 7, here is only useful to know that vertical supports were used much less frequently by the parrots
Line 284-292: I don’t understand the reasoning behind presenting this info at this point of the paragraph, it should go at the beginning where you introduce Table 3 (line 241 and following). Again, the numbers are just repetition of the Table 3, readers only wish to know the most important info, for raw numbers one can refer to the tables
Line 296-297: ok, now I see what the authors were referring to in lines 256-263. I still think the section 256-263 needs some rephrasing
Line 298-299: this info should be presented earlier in the paragraph, around line 245. How much of this result is influenced by the unequal sampling in January?
Line 300-320: all the numbers here are a repetition of data reported in Table 5. You could include sample size in the table as well. Anyway, what all this info mean, what is the main message for the reader? What is the most frequent behaviour and why it is important to know?
Line 320-322: this is the first useful info so far, one this is meaningful and worth reporting and discussing. All the previous info can be much shortened with numbers only provided via tables. Delete “Thus” at the end of line 322
Line 323-346: numbers are repetitions from Table 4 and 7
Figure 2: colour impaired readers might find colours in figure 2c and 2d particularly hard to distinguish. I think this figure is somewhat a repetition of Table 5 and 7, but better summarise the data, thus making the tables redundant
Figure 3 and 4: similar considerations as for Figure 2. Please, use alternative colour palettes to account for colour impaired readers. Again, this figure is somewhat a repetition of all the info reported in the tables, but better summarise the results and is much easier to read. I think the tables would be better reported as supplementary material
Discussion
Line 395-396: too much jargon, not all readers will understand what you are referring to
Line 422: change with “..parrots rejects the idea OF a possible neuromuscular..”
Line 425: perhaps they tend to increase speed when attracted to something of interests (other individual, food)? Or perhaps, the rarity of the observations might indicate injured individuals?
Line 435: change with “…that the use of sidling…”
Line 491-492: I appreciate the effort in addressing the possible limitations of a scan sampling, however could you please expand on how they “do not detract from the overall findings and interpretations”?
Line 498-499: delete “the first quantitative analysis of positional behaviors outside of Mammalia”, it has already been stated in multiple parts of the manuscript
Author Response
We thank the reviewer for the wonderful comments. We believe these revisions have greatly improved the quality of the manuscript. We have addressed most of these suggestions. We are especially thankful for the suggestion to move the tables as supplemental.

Round 2
Reviewer 4 Report
I am glad that the authors found my previous comments useful and adopted some of the changes I suggested. The readability of the manuscript definitely improved by removing some of the tables.
I still have a minor comment in relation to a discrepancy between the numbers presented in the text and in the table in the result section, see below.
Introduction
Line 103: delete “While”, it should read “Any study of positional behaviour is inherently descriptive….”
Line 105-106: I understand you provide references in the predictions, but I still feel that the wording could be improved. Perhaps the sentence should just read “We have developed the following major predictions:” and then provide the predictions.
Methods
Line 184: it would be good to also present the mean number of individuals for each scan together with the range.
Results
Lines 243-250: The authors wrote in response to my comment that the numbers and proportion reported in the text refers to “non-aerial activities outside the nest”. However, it is still unclear how these numbers and proportions were calculated. The reference is to Table S3, in which there are no values for flight and hover, therefore the number of observations and proportions should match what is reported in the text. However, the values in the table for Nest is 3776 and 44.76%, but the authors wrote 3591 and 42.57% in the text. Arboreal locomotion, which is non-aerial activity outside the nest, is 3424 and 40.6% in the table but 3680 and 43.62% in the text. Similarly, all the other values do not match what’s written in Table S3. So, either the numbers should be corrected (in the text or in the table) or the text should be edited to clearly state what these numbers are.
Discussion
Line 469: correct “aquatinted” with “acquainted”
Line 471: correct with “..about HOW Quaker Parrots behave…”
Author Response
Thank you so much for the review
